

# Air mass factor calculation using deep neural network technique for tropospheric NO$_2$ retrieval from space

Yajun Xu[1], Tomohiro O. Sato[1], Ayano Nakamura[1], Tamaki Fujinawa[2,1], Suyun Wang[1], and Yasuko Kasai[3,1]

[1]Terahertz Technology Research Center, National Institute of Information and Communications Technology
[2]Earth System Division, Global Atmospheric Chemistry Section, National Institute for Environmental Studies
[3]Department of Trandiscilinary Science and Engneering, School of Environment and Society, Tokyo Institute of Technology

**Correspondence:** Yasuko Kasai (ykasai@nict.go.jp)

**Abstract.** We performed a feasibility study on using deep neural network (DNN) techniques to calculate the air mass factor (AMF) for the satellite remote sensing of tropospheric nitrogen dioxide (NO$_2$). Satellite remote sensing in the UV and visible wavelength ranges is widely used to study the emission and distribution of tropospheric NO$_2$, which is one of the most crucial gases in both climate change and air pollution. One of the largest sources of uncertainty in NO$_2$ satellite re-

trievals is the AMF, especially when enhanced trace gas concentrations are present in the lower troposphere. Computing the AMF is a very time-consuming task that is usually performed using radiative transfer models. In general, the practical use of such models for satellite remote sensing is limited by the available computational power. We constructed two DNN models to calculate the tropospheric AMF (Trop-AMF-Net) and the altitude-dependent box-AMF (Box-AMF-Net). Trop-AMF-Net consists of five multilayer perceptrons and a linear transform. Box-AMF-Net is an encoder-decoder framework that combines

Trop-AMF-Net with a long short-term memory network. We prepared two test datasets, one of which reflects the actual observed NO$_2$ measurement pattern and the other of which assumes a uniform distribution. For the former, we assumed that the NO$_2$ observation was performed by the Global Observing SATellite for Greenhouse Gases and Water Cycle (GOSAT-GW), which is planned for launch in 2024. For this test dataset, Trop-AMF-Net could reproduce the tropospheric AMFs with root-mean-squared percentage errors (RMSPE) of 0.121 % and 0.156 % when the model was trained using the same and uniform

distributions, respectively. The RMSPEs of the box-AMFs in the troposphere predicted by Box-AMF-Net are 0.284 % for the test and training datasets when the observed pattern was used. The computation time for 10,000 samples using a combination of a central processing unit and graphics processing unit is 3.7 seconds. The RMSPE and computation time are approximately 30 times smaller and two times shorter compared to those of the commonly used look-up table and interpolation technique. Our feasibility study also highlights the importance of training the model with a dataset that is consistent with the test use.

# 1 Introduction

Nitrogen dioxide (NO$_2$) is an important trace gas in the troposphere. The anthropogenic production of NO$_x$ (the sum of NO and NO$_2$) by fossil fuel combustion results in smog and poor air quality. Tropospheric NO$_2$ is a robust tracer of anthropogenic greenhouse gas emissions owing to its short lifetime (of approximately a few hours) (Fujinawa et al., 2021). Enhanced levels





of tropospheric $NO_2$ have been associated with adverse health effects such as cardiovascular and respiratory diseases (e.g.,
Brunekreef and Holgate, 2002; Seposo et al., 2020). In the WHO (2021) guidelines, it is recommended that the annual $NO_2$
dosage should not exceed $60\,\mu g/m^3$.

Satellite remote sensing in the UV and visible spectral ranges is commonly used to measure tropospheric $NO_2$ concentration.
The vertical column density (VCD) of $NO_2$ is obtained by converting the slant column density (SCD) derived from the mea-
sured spectra using an air mass factor (AMF). The differential optical absorption spectroscopy (DOAS) method (Platt et al.,
2008) is widely used to retrieve the SCD of $NO_2$ in satellite remote sensing systems such as the Global Ozone Monitoring
Experiment (GOME) series (Martin et al., 2002; Richter et al., 2011), Ozone Monitoring Instrument (OMI) (Goldberg et al.,
2017), and Tropospheric Monitoring Instrument (TROPOMI) (Van Geffen et al., 2022). Boersma et al. (2004) performed a pre-
cise analysis of the errors incurred in tropospheric $NO_2$ retrieval based on the DOAS method. They showed that the dominant
error source is uncertainty in the AMF. 48–83 % of the VCD error in heavily polluted conditions can be attributed to AMF
uncertainty.

The AMF is the length of the mean light path for photons of a certain wavelength relative to the vertical path. It accounts
for the interactions between emission, absorption, and scattering along the light path. Accurate AMF calculations require the
use of a radiative transfer model (RTM) such as DAK (Stammes, 2001), McArtim (Deutschmann et al., 2011), SCIATRAN
(Rozanov et al., 2002), or VLIDORT (Spurr, 2006). The RTM generally incurs a high computational cost during operational
data processing in satellite remote sensing. Thus, a method that can perform AMF calculations with both high accuracy and
speed is required.

The Look-Up Table (LUT) method is the most commonly used method for AMF calculations in satellite remote sensing
(e.g., Boersma et al., 2011; Cheng et al., 2019; Yang et al., 2023). The AMF values are precalculated using the RTM for
a wide combination of input variables such as the solar zenith angle (SZA), viewing zenith angle (VZA), relative azimuth
angle (RAA), surface albedo, and terrain height, and stored in the LUT. During the LUT-based AMF calculation process, the
input variables in the LUT are queried and the AMF values interpolated for the sample observation conditions. To produce
more accurate AMF values, a finer sampling of the input variables is required in the LUT because of the nonlinearity of
the AMF with respect to the input variables. Because LUT-based AMF calculation involves a tradeoff between accuracy and
computational cost, the selection of the minimum and optimal sets of input variable points (nodes and anchors) to be included
in the LUT is required. Although several mathematical algorithms to optimize the LUT have been proposed (Martino et al.,
2017; Servera et al., 2018), LUT-based interpolation still results in a significant uncertainty of approximately 14 % in AMF
calculations in polluted regions (Lorente et al., 2017).

With the success of machine learning in areas such as computer vision and natural language processing, researchers have
attempted to use machine learning techniques to replace and accelerate RTM-based AMF calculations. Loyola (1999) demon-
strated an AMF calculation with an error of less than 1 % in the SZA range of 0° to 80° that was achieved by applying an
artificial neural network. AMF calculations with improved accuracy based on support vector machines (SVM) have also been
reported (Han et al., 2009). Nikitin et al. (2020) demonstrated the usefulness of neural network and random forest models for
calculating not only the AMF but also cloud and aerosol properties.





In this study, two deep neural network (DNN) models for calculating the tropospheric AMF and altitude-dependent AMF,
hereafter referred to as box-AMF, are proposed as alternatives to the LUT-based method. The datasets used in this study are
described in Sect. 2 and the proposed neural networks for predicting the tropospheric AMF and box-AMF in Sect. 3 The
experimental results are analyzed in Sect. 4 and the conclusions of this study provided in Sect. 5.

## 2 Data description

### 2.1 Calculation of AMF using radiative transfer model

In this study, we used the SCIATRAN v4.6.1 RTM (Rozanov et al., 2017) to calculate the AMF. The tropospheric AMF was
calculated as the sum of the box-AMF for each altitude layer from the surface to the tropopause weighted by the partial vertical
column of trace gas under the assumption of an optically thin trace gas (e.g., Lorente et al., 2017). A correction for the effects
of the temperature on the cross-section was applied to each layer in the AMF calculation.

$$M = \frac{\sum_l m[l]\,(\mathbf{b})\,N_\mathrm{a}[l]c[l]}{\sum_l N_\mathrm{a}[l]}, \tag{1}$$

where $M$ is the tropospheric AMF, $m[l]$ the box-AMF, $N_\mathrm{a}[l]$ the sub-column of the trace gas $NO_2$ in this study, and $c[l]$
the temperature correction factor in the $l$-th altitude layer. $\mathbf{b}$ is the vector of model parameters used in the calculation. The
box-AMF was calculated as

$$m[l] = -\frac{W[l]}{I\sigma\Delta h[l]}, \tag{2}$$

where $W[l]$ is the weighting function at the $l$th altitude layer, $I$ the radiance at the top of the atmosphere (100 km in this study),
$\sigma$ the absorber cross section, and $\Delta h[l]$ the thickness of the altitude layer.

We assumed clear-sky conditions with no clouds or aerosols. The wavelength was set to 440 nm, which is appropriate for
$NO_2$ retrieval (Lorente et al., 2017). The atmospheric profiles for the pressure, temperature, and trace gases were obtained from
climatology datasets prepared using the SCIATRAN software package for 45°N latitude zonal mean in April. Figure 1 shows
the vertical profiles of the pressure, temperature, and $NO_2$ concentration. The vertical resolution is 0.2 km from the surface to
10 km, 0.5 km from 10 to 12 km, and 1 km from 12 to 100 km, i.e., 144 layers were used in this study.

### 2.2 Preparation of datasets

In general, the accuracy of machine learning models is significantly affected by the distribution of the training and test datasets
(e.g., Ben-David et al., 2010). To create a useful machine learning model, the distribution of the training dataset should be
identical to that of the test dataset, which reflects the actual use case.

In our experiments, we prepared two input variable distributions for the training and test datasets (see Table 1). Distribu-
tion A reflects the pattern of the actual observations, whereas distribution B is uniform. The latitude and season conditions



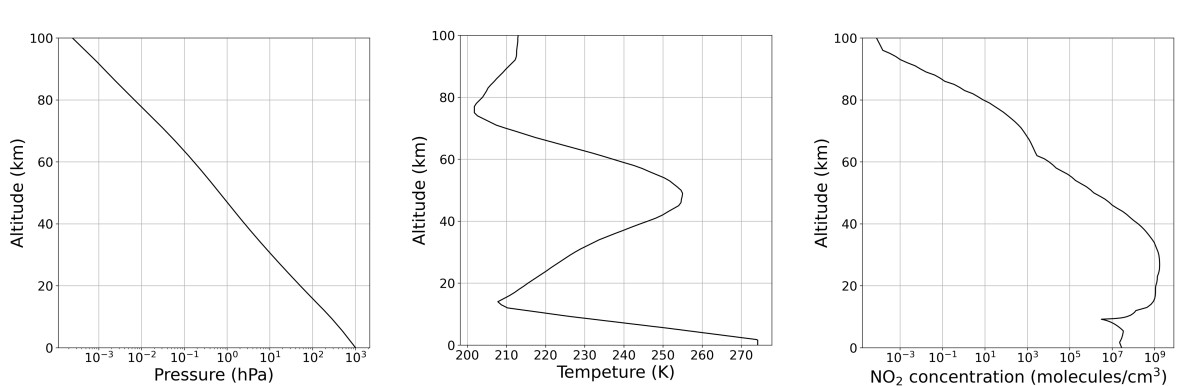

**Figure 1.** Vertical profiles of pressure, temperature, and NO$_2$ concentration used in our radiative transfer calculation.

**Table 1.** Details of the two input variable distributions. $\mu$ represents the mean and $\sigma$ the standard deviation of the normal distribution.

| Input variable | Range | Distribution A | Distribution B |
|---|---|---|---|
| SZA | 0 to 70 (deg) | Normal ($\mu = 39.5$, $\sigma = 4.1$) | Uniform |
| VZA | 0 to 60 (deg) | Uniform | Uniform |
| RAA | 0 to 180 (deg) | 50 %: Normal ($\mu = 52.4$, $\sigma = 5.4$) and 50 %: Normal ($\mu = 127.2$, $\sigma = 5.4$) | Uniform |
| Surface albedo | 0 to 1 | Normal ($\mu = 0.05$, $\sigma = 0.01$) | Uniform |
| Terrain height | 0 to 8 (km) | 43 %: 0 km and 57 %: Uniform | Uniform |

were assumed to be consistent with those of the RTM calculation, that is, a zonal area from 40°N to 50°N latitude in April. To determine the SZA, VZA, and RAA distributions, we assumed that the NO$_2$ observations were performed by the Total Anthropogenic and Natural Emissions Mapping SpectrOmeter-3 (TANSO-3) sensor onboard the Global Observing SATellite for Greenhouse Gases and Water Cycle (GOSAT-GW) satellite, which is planned to be launched in 2024 (Tanimoto et al., 2021). The surface albedo distribution was obtained from the OMI Earth Surface Reflectance Climatology Product OMLER (Kleipool, 2010). The terrain height distribution is a combination of a fixed value of 0 km for the ocean and a uniform distribution for land. The ratios of ocean and land are 43 % and 57 %, respectively, as derived from data from the Advanced Land Observing Satellite (ALOS) project (AW3D30 v4.0) (e.g., Takaku et al., 2021). Distribution A is illustrated in Fig. 2.

The training and test datasets were prepared following Distributions A and B. Hereafter, we distinguish the training and test datasets by uppercase and lowercase letters, respectively; for example, the training dataset based on distribution A is denoted as training dataset A, and the test dataset based on distribution B is denoted as test dataset b. The numbers of samples in the training and test datasets are 100,000 and 10,000, respectively. Figure 3 shows the histograms of the tropospheric AMFs and the vertical distributions of the box-AMFs for the datasets used in this study.



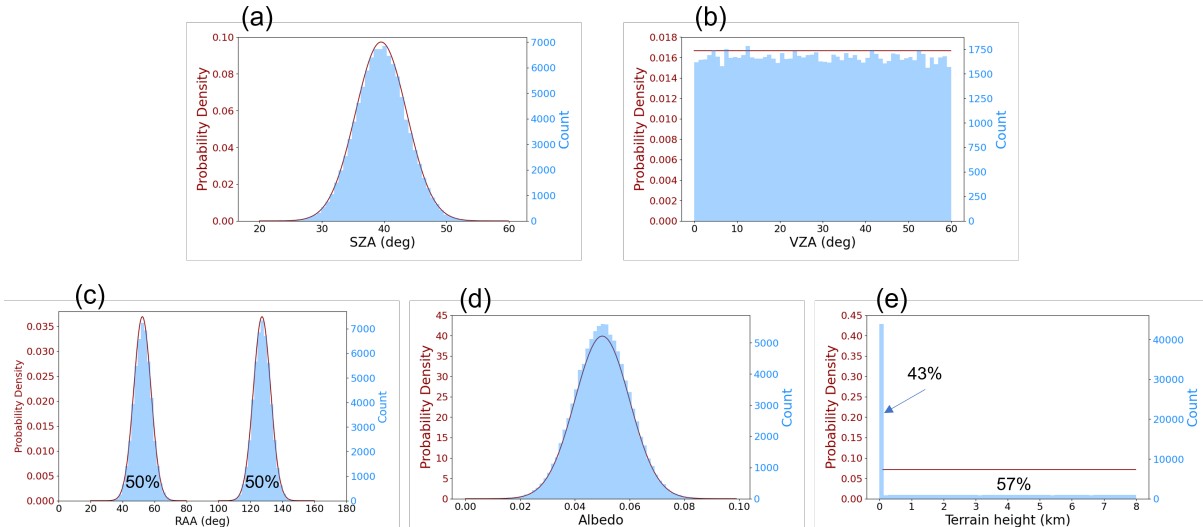

**Figure 2.** Visualization of input variables in distribution A: (a) SZA, (b) VZA, (c) RAA, (d) surface albedo, and (e) terrain height. The details of each distribution are listed in Table 1. The left vertical axis shows the probability density in brown, and the right one the counts in training dataset A (total of 100,000 counts).

**Table 2.** LUT parameter node settings. These are denoted as dataset C, which contains 100,000 samples.

| Input variable | Number of node | Value of node |
| --- | --- | --- |
| SZA (deg) | 10 | 0, 7.78, 15.56, 23.33, 31.11, 38.89, 46.67, 54.44, 62.22, 70 |
| VZA (deg) | 10 | 0, 6.67, 13.33, 20, 26.67, 33.33, 40, 46.67, 53.33, 60 |
| RAA (deg) | 10 | 0, 20, 40, 60, 80, 100, 120, 140, 160, 180 |
| Surface albedo | 10 | 0, 0.11, 0.22, 0.33, 0.44, 0.56, 0.67, 0.78, 0.89, 1 |
| Terrain height | 10 | 0, 0.89, 1.78, 2.67, 3.56, 4.44, 5.33, 6.22, 7.11, 8 |

100    To compare our DNN-based method with the generally used LUT-based method, we prepared the LUT as follows: Linear interpolation was performed to predict the AMF value using the AMF values stored in the LUT of neighborhood nodes of the input variables. The interpolation was performed using regular grid interpolation (RGI) in the scipy module of python (v1.7.3) (Weiser and Zarantonello, 1988). The nodes of the input variables were set at equal intervals, as listed in Table 2. The number of samples in the LUT is consistent with that in the training datasets A and B. We also used this LUT as the training dataset
105    for our DNN-based models and refer to it as training dataset C.





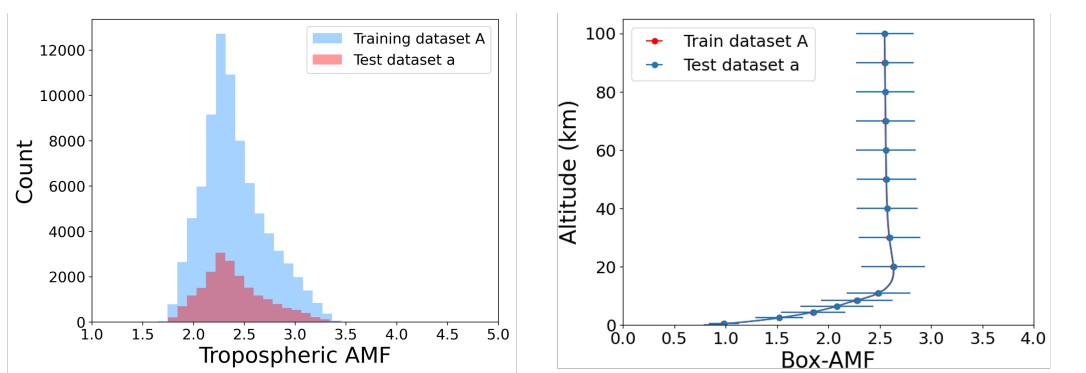

(a) Distribution of tropospheric AMF and Box-AMF of Training dataset A and Test dataset a

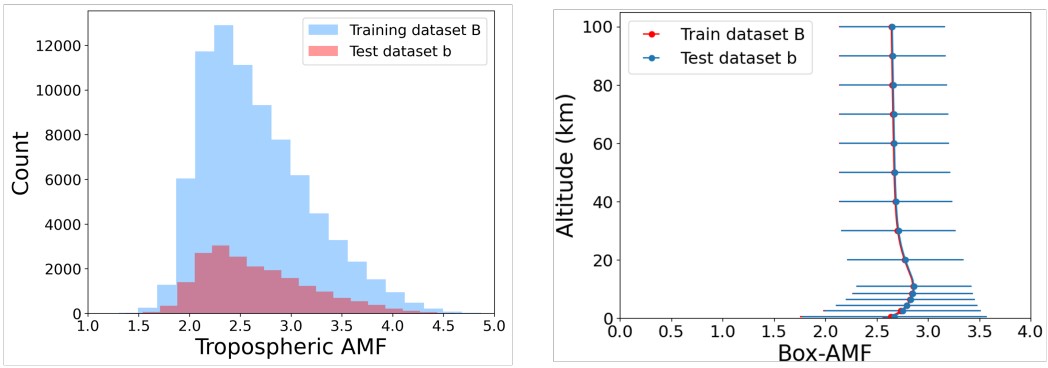

(b) Distribution of tropospheric AMF and Box-AMF of Training dataset B and Test dataset b

**Figure 3.** Tropospheric AMF histograms and vertical distributions of box-AMFs for the training dataset A and test dataset a (a) and the training dataset B and test dataset b (b). In the right box-AMF panels, the mean and standard deviation values are denoted by symbols and error bars, respectively.

## 3 Model description

In this study, we constructed two DNN models to calculate the tropospheric AMF (Trop-AMF-Net) and box-AMFs (Box-AMF-Net) by leveraging the capability of neural networks to capture implicit relationships between input variables and outputs. We set the tropopause to 14 km. The tropospheric AMF was calculated as the sum of box-AMFs from 0 to 14 km following Eq. (1). The box-AMF is a sequence of altitude layers from 0 to 100 km. Thus, we constructed Box-AMF-Net as an encoder-decoder framework that combines Trop-AMF-Net with a long short-term memory (LSTM) network. The input variables in this study are the SZA, VZA, RAA, surface albedo, and terrain height (see details in Sect. 2.2). These input variables are common to both Trop-AMF-Net and Box-AMF-Net.



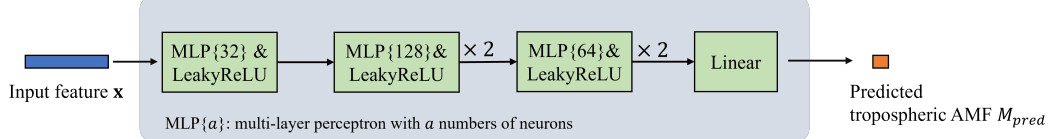

**Figure 4.** Structure of Trop-AMF-Net. Five multi-layer perceptions (MLPs) are applied followed by a LeakReLU activation and linear layer in this model.

## 3.1 Trop-AMF-Net

Figure 4 shows the construction of Trop-AMF-Net. Trop-AMF-Net consists of five multilayer perceptrons (MLPs), each of which is activated by LeakyReLU. The MLPs process the features of the input variable ($\mathbf{x}$). The final layer of Trop-AMF-Net is a linear transform that outputs the predicted tropospheric AMF ($M_{\text{pred}}$) without an activation function. The ground truth for the tropospheric AMF ($M_{\text{truth}}$) was calculated using SCIATRAN with the settings described in Sect. 2.1. The Smooth L1 loss function was employed to minimize the absolute error $L_{AE}$ between the predicted AMF $\hat{y}$ and the ground truth AMF $y$ because of its robustness against outliers (Wang et al., 2020):

$$L_{AE} = \text{smooth}_{L_1}(\hat{y} - y) \tag{3}$$

in which

$$\text{smooth}_{L_1}(a) = \begin{cases} 0.5\frac{a^2}{\delta}, & \text{if } |a| < \delta \\ |a| - 0.5\delta, & \text{otherwise,} \end{cases} \tag{4}$$

$\delta$ was set to 0.01 in this experiment. In addition, considering the variation range of the AMF values (approximately 0–5), the percentage absolute error value was also introduced into the loss function, which is not sensitive to the magnitudes of the numbers in the data. The absolute percentage error $L_{APE}$ is defined as

$$L_{APE} = \frac{|\hat{y} - y|}{y}. \tag{5}$$

The loss for supervising AMF-Net therefore contains two terms:

$$L_{AMF} = L_{AE} + \alpha L_{APE}, \tag{6}$$

where $\alpha$ is a constant introduced to ensure that the two terms have approximately equal weights. We set $\alpha$ to 0.1.

## 3.2 Box-AMF-Net

In general, the box-AMF does not change discontinuously with the altitude. In our RTM calculation, the altitude grid was fixed, and the box-AMFs of the altitude layers below the terrain height defined in the input variables were not calculated.



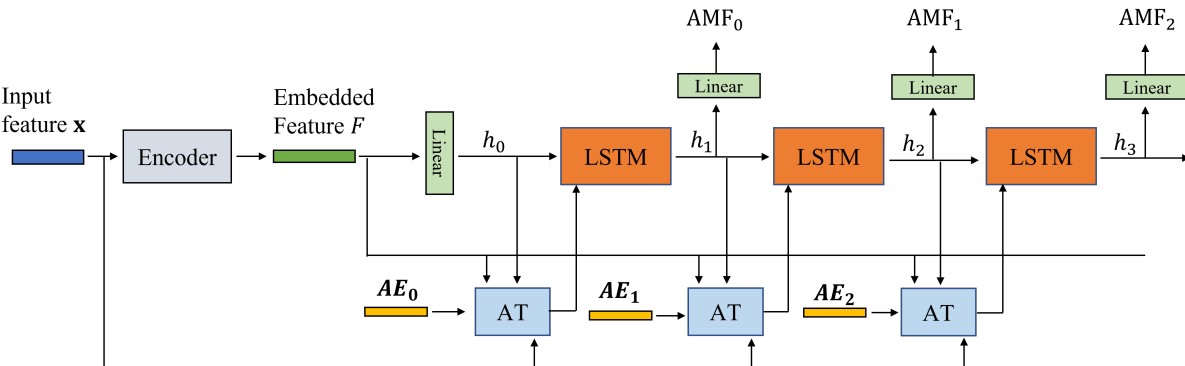

**Figure 5.** Structure of Box-AMF-Net. Box-AMF-Net is an extension of Trop-AMF-Net. The first five layers of Trop-AMF-Net are used as the encoder to extract the embedded feature $F$, and LSTM is used as the decoder to predict the box-AMFs of each air layer. The LSTM hidden state $h_i$, encoded altitude $AE_l$, original input feature $\mathbf{x}$, and embedded features $F$ are fused in the attention block (AT). The box-AMF is predicted by a linear layer using the LSTM hidden state.

That is, the length of the box-AMF vector was not fixed and varied with the terrain height. The box-AMF calculation was

135 therefore assumed to be a prediction problem for variable-length sequences to exploit the advantages of LSTM networks in processing sequential data. LSTM networks have been proven to be powerful tools for modeling sequential data and have contributed significantly to natural language processing, speech recognition, machine translation, and other sequential data analysis problems (Hochreiter and Schmidhuber, 1997).

In Box-AMF-Net, the box-AMF is predicted in a downward direction from the top of the atmosphere (100 km) to the height

140 of the terrain surface. Because the box-AMF varies in an approximately linear manner in the upper atmosphere (Fig. 3, we additionally computed box-AMFs for altitudes up to 120 km with a 1 km resolution using linear extrapolation during the training phase.

Figure 5 shows the construction of Box-AMF-Net, in which the box-AMF is predicted using an encoder-decoder framework. Trop-AMF-Net without the last layer is used as the encoder for extracting the embedded feature $F$ from the input variables $\mathbf{x}$

145 and a LSTM network used as the decoder. The box-AMF of the $l$th altitude layer $m[l]$ is generated from the embedded feature $F$, encoded altitude information (described in Sect. 3.2.1), and hidden state $h_{i-1}$ of the previous LSTM.

### 3.2.1 Altitude Encoding

To account for the correlation between the box-AMFs and altitude, we introduced an altitude encoder proposed by Vaswani et al. (2017). In this method, the altitude encoding of the $l$th altitude layer $AE[l]$ is given by the sine and cosine functions:

150 $$\mathrm{AE}[l, 2d] \quad = \quad sin(l/10000^{2d/D}) \tag{7}$$
$$\mathrm{AE}[l, 2d+1] \quad = \quad cos(l/10000^{2d/D}), \tag{8}$$
$$\tag{9}$$





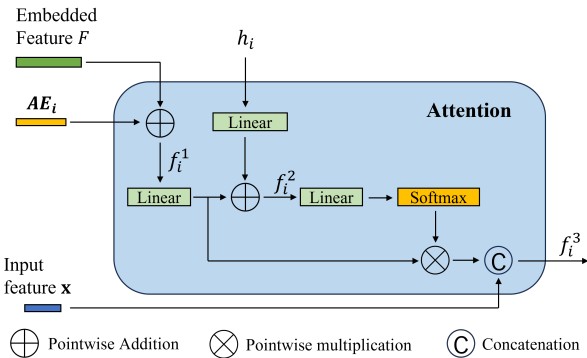

**Figure 6.** Structure of attention block. The embedded feature $F$, input feature $\mathbf{x}$, encoded altitude $\mathrm{AE}[l]$, and hidden state $h_i$ are integrated in this block. The generated feature $f_i^3$ is fed into the LSTM at the next step.

where $l \in [0, 1, ..., L]$ denotes the index of the altitude layer, $D$ the vector dimension of the embedded feature, and $d \in [0, \frac{D-1}{2})$ the index value of each dimension of AE.

### 3.2.2 Attention

Figure 6 shows the detailed structure of the attention block. The encoded altitude of the $l$th altitude layer $\mathrm{AE}[l]$ is fused with the embedded feature $F$ to generate a fused feature $f_i^1$. $f_i^1$ is then passed through a linear layer and fused with the hidden state $h_{i-1}$ of the previous grid to generate $f_i^2$, which is in turn sent to a linear layer followed by softmax activation to generate an attention score. The fused feature $f_i^1$ is weighted by the attention scores and then concatenated with the original physics information $\mathbf{x}$ to generate the final feature $f_i^3$. $f_i^3$ and the current hidden state $h_i$ are transmitted to the LSTM to generate the next hidden state $h_{i+1}$, which is further processed by a linear layer to produce the box-AMF $m[l]$ of the current ($l$-th) layer. The first hidden state $h_0$ is generated from the embedded feature $F$.

### 3.2.3 Loss function

The mean absolute error and mean absolute percentage error were used to supervise the training of Box-AMF-Net. In general, because of the significant variance of the box-AMF in the troposphere, as shown in Fig. 3, we added an altitude-dependent weight $w[l]$ to make Box-AMF-Net more focused during the prediction of the box-AMF in the troposphere.

$$L_{\text{Box-AMF}} = L_{\text{MAE}} + \beta L_{\text{MAPE}} \tag{10}$$

The symbol $\beta$ is a constant introduced so that the weights of the two terms are of equal orders. We set $\beta$ to 0.1. $L_{\text{MAE}}$ is defined as

$$L_{\text{MAE}} = \frac{1}{L} \sum_{l=1}^{L} w[l] * \text{smooth}_{L_1}(m_{\text{pred}}[l] - m_{\text{truth}}[l]), \tag{11}$$



where $L$ is the number of altitude layers, which was set to 144. The altitude-dependent weight $w[l]$ is given by

$$w[l] = \frac{l}{L}, \quad l \in [1, 2, 3..., L].$$
(12)

Note that the altitude layer is numbered from the top of the atmosphere, that is, a smaller $l$ indicates a higher altitude. $L_{\mathrm{MAPE}}$ is defined as

$$L_{\mathrm{MAPE}} = \frac{1}{L} \sum_{l=1}^{L} (w[l] \frac{|m_{\mathrm{pred}}[l] - m_{\mathrm{truth}}[l]|}{m_{\mathrm{truth}}[l]}).$$
(13)

## 4 Experiment

### 4.1 Experimental settings

Both the Trop-AMF-Net and Box-AMF-Net models were implemented using the PyTorch framework and trained using the Adam optimizer with a batch size of 32. Twenty % of the training dataset was used as a validation dataset during training. The
samples in the validation dataset were randomly selected. The learning rate was initially set to 0.001 and decreased by 80% if the loss of the validation dataset did not decrease for eight epochs. The training process was stopped when the loss did not decrease for 20 epochs.

To compare the computation times of our DNN-based model and the LUT model, we used an Intel(R) Xeon(R) Silver 4210 central processing unit (CPU) @2.20GHz and a NVIDIA Quadro P400 graphics processing unit (GPU).

### 4.2 Trop-AMF-Net Results

Tropospheric AMF values were calculated using Trop-AMF-Net and the LUT for the two test datasets (a and b). Three training datasets (A, B, and C) were used for Trop-AMF-Net. Table 3 summarizes the tropospheric AMF predictions. The evaluation metrics used are the root-mean-square error (RMSE), root-mean-square percentage error (RMSPE), and coefficient of determination ($R^2$).

$$\mathrm{RMSE} = \sqrt{\frac{1}{N} \sum_{i=1}^{N} (M_{\mathrm{pred}}[i] - M_{\mathrm{truth}}[i])^2},$$
(14)

$$\mathrm{RMSPE} = \sqrt{\frac{1}{N} \sum_{i=1}^{N} \left(\frac{M_{\mathrm{pred}}[i] - M_{\mathrm{truth}}[i]}{M_{\mathrm{truth}}[i]}\right)^2},$$
(15)

$$R^2 = 1 - \frac{\sum_{i=1}^{N} (M_{\mathrm{truth}}[i] - M_{\mathrm{pred}}[i])^2}{\sum_{i=1}^{N} (M_{\mathrm{truth}}[i] - \bar{M}_{\mathrm{truth}})^2}, \quad \bar{M}_{\mathrm{truth}} = \frac{1}{N} \sum_{i=1}^{N} M_{\mathrm{truth}}[i]$$
(16)

where $N$ is the total number of samples.

In the prediction on test dataset a, the best prediction was achieved by Trop-AMF-NET trained on training dataset A (No.
1a) with an RMSE of 0.003, RMSPE of 0.121 %, and $R^2$ of 0.99992. The second and third best predictions were achieved by





**Table 3.** Summary of AMF predictions using Trop-AMF-Net and the LUT for the two test datasets. The test dataset a is based on the distribution A, which reflects the actual observed pattern described in Sect. 2.2. The test dataset b is based on distribution B, which is a uniform distribution. The Trop-AMF-Net model was trained using three different datasets A, B, and C based on distributions A and B and the LUT, respectively.

| No. | Test dataset | Method | Training datset | RMSE | RMSPE [%] | $R^2$ |
|-----|-----|-----|-----|-----|-----|-----|
| 1a | | | A (observation) | 0.003 | 0.121 | 0.99992 |
| 1b | | Trop-AMF-Net | B (uniform) | 0.004 | 0.156 | 0.99987 |
| 1c | a (observation) | | C (LUT) | 0.027 | 1.277 | 0.99260 |
| 1d | | LUT | - | 0.046 | 2.302 | 0.97825 |
| 2a | | | A (observation) | 0.425 | 16.965 | 0.37691 |
| 2b | | Trop-AMF-Net | B (uniform) | 0.004 | 0.144 | 0.99995 |
| 2c | b (uniform) | | C (LUT) | 0.013 | 0.617 | 0.99941 |
| 2d | | LUT | - | 0.022 | 0.949 | 0.99840 |

Trop-Net-AMF trained on training datasets B (No. 1b) and C (No. 1c), respectively. The best RMSPE is 0.121 % (No. 1a), which is approximately 10 times smaller than 1.277 % (No. 1c). The RMSPE of the LUT method is 2.302 % (No. 1d), which is approximately 20 times higher than that of the best case (No. 1a). In contrast, the best prediction on test dataset b was achieved by Trop-AMF-NET trained on training dataset B (No. 2b) with an RMSE of 0.004, RMSPE of 0.144 %, and $R^2$ of 0.99995. Trop-AMF-Net trained on training dataset A (No. 2a) provided the worst predictions. Its RMSPE of 16.965 % is approximately 100 times higher than that of the best case (No. 2b). These results clearly show the importance of consistency between the test and training datasets for Trop-AMF-Net and the risks of imposing limitations on the generalizability of the network by relying on data from a specific distribution.

The difference between the Trop-AMF-Net and LUT methods are discussed by comparing the performance of Trop-AMF-Net trained using dataset C and the LUT method. The RMSPE values are 1.277 % (No. 1c) and 2.302 % (No. 1d) for test dataset a and 0.617 % (No. 2c) and 0.949 % (No. 2d) for test dataset b. Compared to the LUT method, the RMSPE values of Trop-AMF-Net are smaller by approximately 30—50 %.

Figure 7 shows a comparison of the tropospheric AMF values predicted by Trop-AMF-Net and the ground truth for the No. 1a, 1b, 2a, and 2b cases. The results show that Trop-AMF-Net could replicate the ground truth values for the No. 1a, 1b, and 2b cases. The prediction errors are relatively larger at the two ends of the ground truth values, at which there are only small numbers of samples. However, the error in case No. 2a case is larger than those of the other cases because the training dataset did not cover the entire range of the test dataset. Training Dataset A is based on the assumed observation pattern in which the input variables are concentrated within certain ranges, as shown in Fig. 2. This reduced the number of input variable combinations that the model could cover and led to larger errors.







**Figure 7.** Comparison between tropospheric AMF values predicted by Trop-AMF-Net and the ground truth for the No. 1a, 1b, 2a, and 2b cases. The left column is the scatter plot of the predicted ground truth values. The middle and right columns show the RMSE and RMSPE as functions of the ground truth value in brown color. The blue bars are the ground truth histograms.



**Table 4.** Comparison of computation time. The total time to process 10,000 samples was measured.

|  | Trop-AMF-Net (w/ GPU) | Trop-AMF-Net (w/o GPU) | LUT | RTM |
|---|---|---|---|---|
| Time [s] | 0.23 | 0.75 | 3.8 | $4\times10^4$ |

Table  4 lists the computation times for processing 10,000 samples from the test datasets. Trop-AMF-Net was run on both the CPU and GPU and on only the CPU. The LUT and RTM calculations were performed using only the CPU. The fastest method is Trop-AMF-Net executed on the CPU and GPU, which required a running time of 0.23 s. The time required by the LUT-based method is 3.8  s. RTM required a much longer time ($4\times10^4$  s). These results demonstrate that Trop-AMF-Net can reduce the computation time by approximately 17 times compared to the LUT-based method by using both a CPU and GPU.

**4.3  Box-AMF-Net Results**

The box-AMF values were calculated using Box-AMF-Net and the LUT for two test datasets (a and b) and three training datasets (A, B, and C) to evaluate Trop-AMF-Net, which was described in Sect.  4.2. Table  5 summarizes the box-AMF prediction results. The evaluation metrics are the RMSE, RMSPE, and $R^2$ averaged over the altitude layers.

$$\text{Averaged RMSE} \quad = \quad \frac{1}{N}\sum_{i=1}^{N}\sqrt{\frac{1}{L}\sum_{l=1}^{L}\left(M_{\text{pred}}[i,l]-M_{\text{truth}}[i,l]\right)^2} \tag{17}$$

$$\text{Averaged RMSPE} \quad = \quad \frac{1}{N}\sum_{i=1}^{N}\sqrt{\frac{1}{L}\sum_{l=1}^{L}\left(\frac{M_{\text{pred}}[i,l]-M_{\text{truth}}[i,l]}{M_{\text{truth}}[i,l]}\right)^2} \tag{18}$$

$$\text{Averaged R}^2 \quad = \quad \frac{1}{N}\sum_{i=1}^{N}\left(1-\frac{\sum_{l=1}^{L}\left(m_{\text{truth}}[i,l]-m_{\text{pred}}[i,l]\right)^2}{\sum_{l=1}^{L}\left(m_{\text{truth}}[i,l]-\bar{m}_{\text{truth}}[l]\right)^2}\right), \quad \bar{m}_{\text{truth}}[l]=\frac{1}{N}\sum_{i=1}^{N}m_{\text{truth}}[i,l] \tag{19}$$

Here, $N$ is the total number of samples and $L$ the number of altitude layers.

Box-AMF-Net exhibited similar behavior to that of Trop-AMF-Net. The best predictions for test datasets a and b were given by Box-AMF-Net trained on datasets with the same distributions as the test datasets, i.e., training datasets A and B in the
No. 1a and 2b cases, respectively. The averaged RMSPE values for all altitude layers (0–100 km) are 0.166 % and 0.155 %, respectively. For all the cases, the average RMSE, RMSPE, and $R^2$ values for the troposphere (0–14 km) are twice as large as those for all the altitudes. The more significant changes in the box-AMF values in the troposphere compared to those in the stratosphere and above make predictions more difficult. The RMSPE values of the No. 1c and 2c cases are approximately 50% smaller than those of the No. 1d and 2d cases. Similar to Trop-AMF-Net, the RMSPE value of Box-AMF-Net is reduced by
approximately 50 % compared to that of the LUT method. The RMSPE values of the No. 2a case for all altitude layers and the troposphere are 7.316 % and 10.826 %, respectively, which are 40–50 % higher than those of the No. 2b case. The risk of limiting the performance of a DNN-based model trained using a dataset based on a specific distribution was also demonstrated by Box-AMF-Net and Trop-AMF-Net.





**Table 5.** Same as Table 3 but for Box-AMF-Net. The averaged RMSE, RMSPE, and R$^2$ values are shown for all the altitude layers (0–100 km) and for the troposphere (0–14 km).

| No. | Test dataset | Method | Training datset | Averaged RMSE | | Averaged RMSPE [%] | | Averaged R$^2$ | |
|---|---|---|---|---|---|---|---|---|---|
| | | | | 0–100 km | 0–14 km | 0–100 km | 0–14 km | 0–100 km | 0–14 km |
| 1a | | | A (observation) | 0.002 | 0.004 | 0.166 | 0.284 | 0.99982 | 0.99956 |
| 1b | a (observation) | Box-AMF-Net | B (uniform) | 0.008 | 0.012 | 0.473 | 0.762 | 0.99764 | 0.99196 |
| 1c | | | C (LUT) | 0.026 | 0.045 | 2.213 | 3.704 | 0.99780 | 0.92419 |
| 1d | | LUT | - | 0.073 | 0.135 | 4.540 | 8.335 | 0.84042 | 0.57270 |
| 2a | | | A (observation) | 0.217 | 0.326 | 7.316 | 10.826 | -0.03720 | -1.70790 |
| 2b | b (uniform) | Box-AMF-Net | B (uniform) | 0.003 | 0.005 | 0.155 | 0.260 | 0.99978 | 0.99946 |
| 2c | | | C (LUT) | 0.007 | 0.011 | 0.401 | 0.710 | 0.99920 | 0.99802 |
| 2d | | LUT | - | 0.018 | 0.024 | 0.866 | 1.272 | 0.99519 | 0.99518 |

**Table 6.** Same as Table 4 but for Box-AMF-Net.

| | Box-AMF-Net (w/ GPU) | Box-AMF-Net (w/o GPU) | LUT | RTM |
|---|---|---|---|---|
| Time [s] | 3.7 | 47 | 6.9 | $4\times10^5$ |

The RMSPE of the prediction results for each altitude layer is shown in Fig. 8. There is a relative increase in the RMSPE values at altitudes close to the ground (particularly below 10 km) for all the methods. This increase is due to the relatively small amount of box-AMF data because of the terrain height and the significant changes in the box-AMFs in this altitude range. For each altitude layer, Box-AMF-Net performed better than the LUT method, except for test dataset b and training dataset A (No. 2a).

Table 6 lists the computation times required to process 10,000 samples of the test datasets. Similar to Trop-AMF-Net, Box-AMF-Net was run on both the CPU and GPU, and run on only the CPU. The LUT and RTM calculations were performed using only the CPU. Box-AMF-Net on the CPU and GPU required the shortest time of 3.7 s, which is approximately twice as fast as the LUT method.

## 4.4 Discussion on Box-AMF-Net

Here, we describe the effects of two schemes employed in the Box-AMF-Net method. The first scheme is the addition of the altitude-dependent weights $w[l]$ defined in Eq. (12). Figure 9 shows the RMSPE values for each altitude layer with and without altitude-dependent weights for test dataset a and training dataset A. The RMSPE values with the weights are 0.1817 %





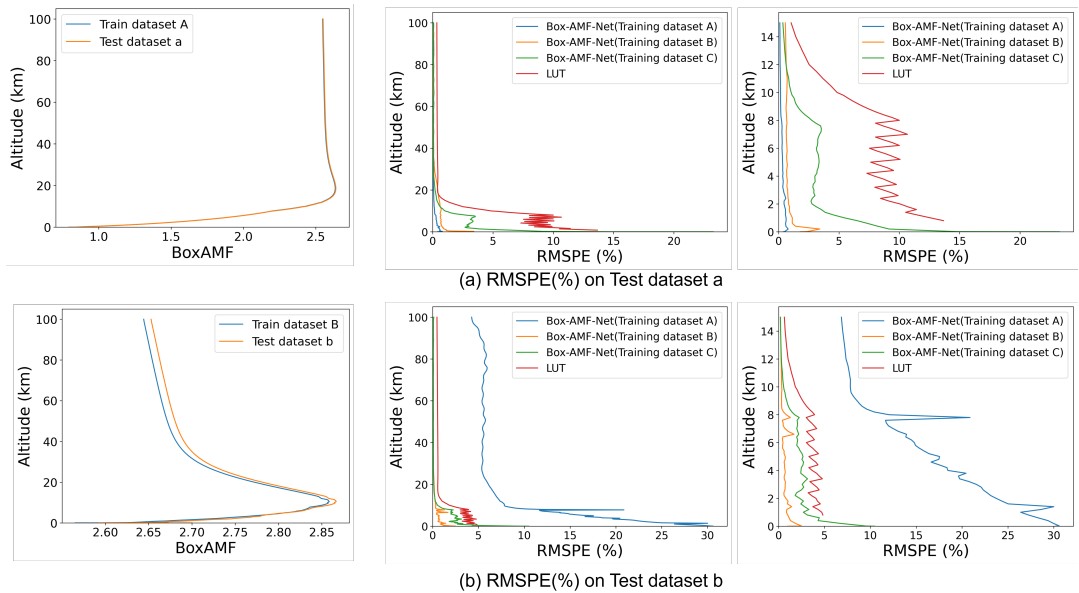

**Figure 8.** Comparison of RMSPE at different altitude layers between Box-AMF-Net and the LUT-based method. (a) is the RMSPE on test dataset a and (b) the RMSPE on test dataset b. The left column shows the box-AMF, and the middle and right ones show the RMSPE values at the altitude ranges of 0–100 km and 0–14 km, respectively.

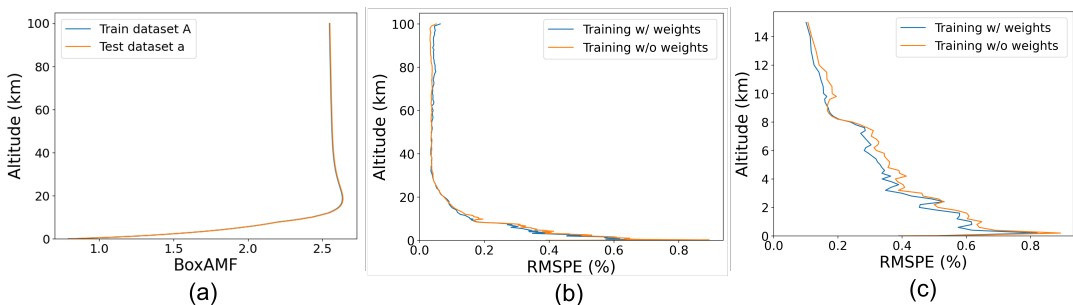

**Figure 9.** Comparison of RMSPE values for Box-AMF-Net trained with and without altitude-dependent weights using test dataset a and training dataset A. Panel (a) shows the box-AMFs and panels (b) and (c) the RMSPE values at the altitude ranges of 0–100 km and 0–14 km, respectively.

and 0.3135 % for 0–100 km and 0–14 km, respectively; those without the weights are 0.1655 % and 0.2838 %, respectively. The RMSPE values with weights are approximately 10 % smaller than those without weights.

The second scheme is the linear extrapolation of the box-AMF for altitudes of up to 120 km in the training phase. Figure 255 10 shows the RMSPE values for each altitude layer with and without linear extrapolation for the case of test dataset a and



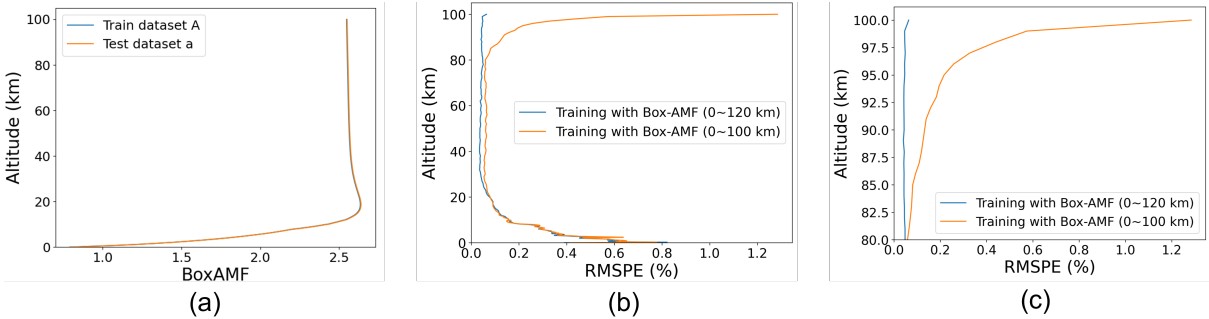

**Figure 10.** Comparison of RMSPE[%] values for each altitude layer with and without linear extrapolation for test dataset a and training dataset A. Panel (a) shows the box-AMFs and panels (b) and (c) the RMSPE values at the altitude ranges of 0–100 km and 80–100 km, respectively.

training dataset A. The RMSPE values were significantly decreased by this linear extrapolation at altitudes above 80 km. In the LSTM method, the generation of the initial state by embedded features rather than self-iterative generation resulted in a larger error at the beginning of the altitude layer (Hochreiter and Schmidhuber, 1997). By adding the extrapolated box-AMF values during the training phase, this deficiency of LSTM was significantly improved. We also note that this extrapolation scheme was not effective for all the cases. The scheme should be applied to sequence data with small variances or changes that are well approximated by a simple equation like the linearly changing box-AMF predictions in this study.

## 5 Conclusions

In this study, a DNN-based approach to calculate AMF values was proposed as an alternative to the LUT-based method. We constructed two DNN models to calculate the tropospheric AMF (Trop-AMF-Net) and altitude-dependent box-AMFs (Box-AMF-Net). The input variables are the SZA, VZA, RAA, surface albedo, and terrain height. Clear-sky conditions without clouds or aerosols were assumed. We prepared two distributions for these input variables comprising distribution A, which reflects the actual observed distribution, and distribution B, which corresponds to a uniform distribution. AMF values were calculated using the SCIATRAN radiative transfer model.

Trop-AMF-Net consists of five multilayer perceptrons and a linear transform that outputs the predicted tropospheric AMF without an activation function. This model can reproduce the tropospheric AMF with a maximum error that is approximately 20 times smaller within a computation time that is approximately 17 times shorter compared to those of the commonly used LUT method. Box-AMF-Net is an encoder-decoder framework with a Trop-AMF-Net encoder based on multilayer perceptrons and a LSTM network decoder. We used two schemes to reduce the error in the box-AMF prediction in this model, namely, the addition of an altitude-dependent weight to the loss function and the linear extrapolation of the data from 100 to 120 km in the



training phase. Our Box-AMF-Net can reproduce the box-AMF values with a maximum error that is approximately 30 times smaller within a computation time that is approximately two times shorter compared to those of the LUT method.

We emphasize the importance of consistency in the distribution between the training and test datasets. Although a training dataset with a uniform distribution resulted in a model with generally good performance, this is not the best achievable performance. The best performance was obtained by using training and test datasets based on the same distributions. However,

this comes with the risk of limiting the generalizability of the network because of reliance on data from a specific distribution. Therefore, we recommend the following: If the use case of the model can be approximated well by a certain distribution, the model should be trained using this distribution. Otherwise, using a training dataset based on a uniform distribution is the better option.

*Author contributions.* Yajun Xu: designed the algorithm, carried out the experiment, and wrote this manuscript. Tomohiro O. Sato: designed
and supervised this study, designed a part of the algorithm, and wrote this manuscript. Ayano Nakamura: designed a part of the algorithm, generated the dataset using RTM, and prepared this manuscript. Tamaki Fujinawa: calculated the SZA, VZA and RAA distributions used in this study, and improved this manuscript. Suyun Wang: designed a part of the algorithm, improved this manuscript. Yasuko Kasai: provided resources and supervised this study.

*Competing interests.* The authors declare that they have no conflict of interest.

*Acknowledgements.* We would like to thank the University of Bremen for sharing the SCIATRAN software with us. The surface albedo data used in this paper were acquired as part of the activities of NASA's Science Mission Directorate, and are archived and distributed by the Goddard Earth Sciences (GES) Data and Information Services Center (DISC). The terrain height data used in this study is based on the free public ALOS data of Japan Aerospace Exploration Agency (JAXA).



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
