# Peer review of "Air mass factor calculation using deep neural network technique for tropospheric NO2 retrieval from space"

_EGUsphere, 2024_

## Referee Comment (RC1)

**Comments**

Manuscript: egusphere-2024-194

The manuscript under review presents a possible novel approach for calculating the air mass factor (AMF) for tropospheric nitrogen dioxide (NO2) retrieval from satellite data using neural networks (NNs). This method diverges from traditional techniques dependent on radiative transfer models and look-up tables, which are typically constrained by computational resources and time.

The authors developed two specific NN models: Trop-AMF-Net, designed to calculate the tropospheric AMF, and Box-AMF-Net, aimed at determining altitude-dependent AMF. Both models have significantly reduced computation time. The manuscript emphasizes the criticality of aligning the distributions of training and testing datasets to optimize model performance, hinting at the limited generalizability of those NNs across varying atmospheric conditions.

The research could have implications for the field of atmospheric science, particularly in enhancing real-time data analysis and environmental monitoring capabilities. However, the necessity for broader validation and testing is rightly noted, emphasizing the need to understand the full scope and limitations of these methodologies comprehensively.

It is suggested that the authors clarify their propositions early in the manuscript. The paper represents initial steps in applying deep learning to the chosen application, providing a clear framework while noting limitations for future studies to address in developing refined systems.

**General Comments**

The manuscript clearly states its limitations regarding clear sky conditions. There is still a challenge in AMF computation related to uncertainties in the aerosol and cloud conditions. Nonetheless, there seems to be an advancement in performance, primarily aimed at enhancing computational efficiency.

The selection of algorithms for training and quality assessment seems correct. The construction of the training/test and validation is clearly depicted. Their choice of the algorithm's architecture is coherent with the task at hand, supported by reasoning.

The authors revisited the literature properly and brought to discussion the kind of measurements they expect to handle and their respective limitations. The detailed discussion surrounding dataset characteristics is particularly beneficial for contextual understanding.

**Specific Comments**

**Generalization and Dataset Diversity:**

The following statement raises concerns about potential overfitting within controlled scenarios:

"Although a training dataset with a uniform distribution resulted in a model with generally good performance, this is not the best achievable performance. The best performance was obtained by using training and test datasets based on the same distributions. However, this comes with the risk of limiting the generalizability of the network because of reliance on data from a specific distribution."

The disparity in performance between matched and mismatched training and test datasets highlights a need for broader generalizability. Future practical applications of these tools may not encounter perfectly aligned data, a factor worth considering in model development.

**Model Interpretability:**
The complex nature of NN approaches to AMF calculations may limit the interpretability compared to conventional methods. While this may not affect certain applications, discussing the limitations in understanding atmospheric interactions through this new approach would be valuable.

**Limiting Assumptions:**
The assumption of clear-sky conditions neglects the significant impact of aerosols and clouds on AMF computations. Given that these elements represent substantial uncertainties, a more in-depth exploration of how these assumptions affect model applicability and performance would be beneficial.

**Technical Corrections**

**Clarification needed (Lines [83-84]):**

"To create a useful machine learning model, the distribution of the training dataset should be identical to that of the test dataset"

This statement should be revisited. While it's ideal for the training and test data to come from similar distributions, strict identity is not necessary. In fact, the reference in the

respective paragraph examines the bound to how different a target domain can be, in comparison to the training data without compromising too much the model performance.

In practice, the training and test data may come from slightly different distributions due to data collection methods, sampling bias, temporal changes, among other factors. I believe this scenario is expected from the problem the authors tackled, due to the nature of the variables and because of the future data collection method from the new sensor mentioned in the manuscript. While it is essential to minimize covariate shift, models can still perform well with non-identical distributions.

The use of machine learning usually aims to create models that generalize well to unseen data. If the model learns relevant patterns from the training data, it should still perform well on the test data, even if the distributions differ slightly. A good model is robust to minor differences.

This manuscript could benefit from a discussion on strategies to enhance model robustness to distribution shifts, such as transfer learning techniques (i.e. fine-tuning) to allow models trained on one distribution to adapt to another, or even the possibility of using nonparametric models, that tend to be more robust to distribution shifts for making fewer assumptions about the data distribution.

---

## Author Comment (AC2)

We deeply appreciate your review. Overfitting is an interesting problem, which we have addressed to some extent in our experiments. Using the Trop-AMF experiment results as a case in point (Tab 3 is conveniently attached for reference), we can see from Experiment No. 2a that the network trained on training dataset A (derived from a real distribution with a limited data range) performs poorly on test dataset b (random distribution with a broader data range), as you pointed out overfitting. However, when we trained the network with training dataset B (random distribution, wide range of data), the network performed well on both test datasets a and b. (cf. Experiments No. 1a and 1b). We plan to improve the manuscript not to be misleading.

Therefore, we recommend training the network a large range and a large amount of random data to, allowing the network to cover the any possible data in order to eliminate uncertainty in unseen scenarios. We would improve our manuscript according to your comments.

**Table 3.** Summary of AMF predictions using Trop-AMF-Net and the LUT for the two test datasets. The test dataset a is based on the distribution A, which reflects the actual observed pattern. The test dataset b is based on distribution B, which is a uniform distribution. The Trop-AMF-Net model was trained using three different datasets A, B, and C based on distributions A and B and the LUT, respectively.

| No. | Test dataset | Method | Training datset | RMSE | RMSPE [%] | $R^2$ |
|-----|-------------|--------|-----------------|------|-----------|-------|
| 1a | | | A (observation) | 0.003 | 0.121 | 0.99992 |
| 1b | a (observation) | Trop-AMF-Net | B (uniform) | 0.004 | 0.156 | 0.99987 |
| 1c | | | C (LUT) | 0.027 | 1.277 | 0.99260 |
| 1d | | LUT | - | 0.046 | 2.302 | 0.97825 |
| 2a | | | A (observation) | 0.425 | 16.965 | 0.37691 |
| 2b | b (uniform) | Trop-AMF-Net | B (uniform) | 0.004 | 0.144 | 0.99995 |
| 2c | | | C (LUT) | 0.013 | 0.617 | 0.99941 |
| 2d | | LUT | - | 0.022 | 0.949 | 0.99840 |